# Early childhood female genital mutilation in Sierra Leone, 2008–2019

**Augustus Osborne** [1]*, **Camilla Bangura**[2], **Baindu Abu**[3], **Umaru Sesay**[4,5]

**1** Institute for Development, Freetown, Western area, Sierra Leone, **2** Department of Biological Sciences, School of Basic Sciences, Njala University, PMB, Freetown, Sierra Leone, **3** Ministry of Health, Western Area, Freetown, Sierra Leone, **4** Sierra Leone Field Epidemiology Training Program, National Public Health Agency, Western Area, Freetown, Sierra Leone, **5** Africa Field Epidemiology Network, Western Area, Freetown, Sierra Leone

* augustusosborne2@gmail.com

## Abstract

### Background

Female genital mutilation remains a public health concern and human rights violation affecting young girls in Sierra Leone, despite global efforts to eliminate the practice. With its diverse socio-cultural background and varying regional practices, Sierra Leone presents a unique context for examining how female genital mutilation practices have evolved across different population subgroups. This study examined the early childhood female genital mutilation among women of reproductive age (15–49 years) who reported having FGM before the age of five in Sierra Leone.

### Methods

The study utilised data from the Sierra Leone Demographic Health Survey rounds conducted in 2008, 2013, and 2019. The World Health Organisation Health Equity Assessment Toolkit software calculated various measures, including difference, ratio, population-attributable risk, and population-attributable fraction. An assessment was calculated for five stratifiers: age, education level, economic status, place of residence, and sub-national province.

### Results

The prevalence of female genital mutilation among women of reproductive age (15–49 years) who reported undergoing the practice before the age of five in Sierra Leone declined from 23.2% in 2008 to 12.3% in 2019. By 2019, female genital mutilation showed minimal variation between women aged 40–49 and 15–19, as well as in urban-rural differences. Economic variations in female genital mutilation decreased but continued to disadvantage women in the poorest quintile. Educational variations in female genital mutilation decreased but still impacted women without formal

**Data availability statement:** Third party data was obtained for this study from The DHS Program. Data may be requested from The DHS Program after creating an account and submitting a concept note. More access information can be found on The DHS Program website (https://dhsprogram.com/data/Access-Instructions.cfm). The authors confirm that interested researchers would be able to access these data in the same manner as the authors. The authors also confirm that they had no special access privileges that others would not have.

**Funding:** The author(s) received no specific funding for this work.

**Competing interests:** The authors have declared that no competing interests exist.

education. Provincial variations in female genital mutilation widened, with the ratio between the Western and Northwestern provinces increasing from 1.8 in 2008 to 2.6 in 2019.

## Conclusion

The results showed a decrease in early childhood female genital mutilationin Sierra Leone. While differences related to age groups and urban-rural residence have largely been eliminated, substantial differences persist across educational levels, economic status, and provinces. Most notably, the provincial differences between the Western and Northwestern provinces had widened, with the difference in ratio indicating that female genital mutilation practices remain disproportionately concentrated in some provincial areas. These results suggest that while national-level interventions have been partially successful, there is a critical need for targeted, context-specific approaches that address persistent socioeconomic and provincial variations to achieve a more equitable reduction in early childhood female genital mutilation across all population groups in Sierra Leone.

## Introduction

Female Genital Mutilation (FGM) encompasses all procedures involving partial or total removal of the external female genitalia or other injury to the female genital organs for non-medical reasons [1]. The World Health Organization classify FGM into four types, including clitoridectomy and infibulation, all of which can lead to severe immediate and long-term health complications [2]. These complications include severe pain, excessive bleeding, infections, complications during childbirth, and psychological trauma [3].

Globally, more than 200 million women and girls have undergone FGM, with an additional 3 million girls at risk annually [2]. The practice is most prevalent in 30 countries across Africa, the Middle East, and Asia [2]. In sub-Saharan Africa, FGM prevalence varies significantly, with over 144 million girls and women being cut [4]. In West Africa, the prevalence of FGM has seen a marked decline of 48.2%, dropping from 73.6% in 1995 to 25.4% by 2017 [5]. Yet, an estimated one-fourth of FGM procedures performed on girls are now being executed by healthcare professionals a rate that doubles for adolescents [6]. In some countries, like Sierra Leone, promoting the involvement of medical practitioners has been framed as a harm-reduction approach. However, this method raises serious ethical issues, as it contradicts medical ethical standards, heightens the risks girls and women face about FGM, and hampers global efforts to achieve the complete abolition of the practice [6].

In Sierra Leone FGM is a deep-rooted cultural practice and traditional belief often performed as part of initiation rites into secret societies, named the Bondo society, making it a complex social and public health challenge [4]. The practice is headed by a traditional leader named "Sowei", who acts as a symbolic guardian, playing a pivotal

role in preserving and honoring the traditional customs of the community [7]. This practice typically extends over several weeks in secluded rural settings to ensure the rituals' integrity and focus remain undisturbed. The most common type of FGM practiced in Sierra Leone is type 2 [8]. The country is categorized as a Group 1, with an FGM prevalence exceeding 80% [9]. Historically, Sierra Leone has reported one of the highest FGM prevalence rates globally, with approximately 83.0% of women aged 15–49 having undergone the procedure and, alarmingly, 13.2% of girls experiencing FGM before age five [8]. Recent trends suggest a slight decline, though the practice remains widespread, particularly in rural areas [9].

Sierra Leone has implemented various initiatives to address FGM, including the Child Rights Act 2007 and the National Strategy for Reducing FGM/C (2016–2020) [9]. However, the absence of explicit legislation criminalising FGM has hampered progress [9]. The government's efforts, supported by international organisations and local NGOs, face significant challenges, including strong cultural resistance, limited resources, and the practice's connection to powerful traditional institutions [9]. The COVID-19 pandemic has further complicated intervention efforts, with reports suggesting an increase in FGM cases during lockdown periods.

Studies across Africa have identified several factors associated with FGM, including educational level, socioeconomic status, religious beliefs, and geographical location [10–17]. Research has demonstrated that maternal education and urban residence are inversely associated with FGM prevalence [18–20]. Additionally, studies have shown that women's empowerment and exposure to anti-FGM messages through mass media can significantly influence attitudes toward the practice [21–23]. However, the complex interplay between these factors varies considerably across different cultural contexts.

While several studies have examined FGM in Sierra Leone, most have focused on prevalence and immediate health consequences [6,9,24–33]. No research exists on early childhood FGM over time, particularly regarding socioeconomic and geographic variations over 11 years (2008–2019) in Sierra Leone. This study aims to fill this crucial knowledge gap by analysing the prevalence and patterns of FGM among women of the reproductive age (15–45 years) who reported undergoing the practice before the age of five years old in Sierra Leone from 2008 to 2019. Understanding these patterns and changes over time is essential for developing targeted interventions and informing policy decisions. The findings will contribute to the existing literature by providing a comprehensive analysis of early childhood FGM prevalence over time and associated differences, thereby supporting evidence-based strategies for FGM prevention and elimination in Sierra Leone.

## Methods

### Study design and source

The study employed a time-trend ecological study design. We utilised data from the Sierra Leone Demographic Health Survey (SLDHS) conducted in 2008, 2013, and 2019. The SLDHS is a national survey designed to uncover differences in social concerns, health indicators, and household demographics [8]. The survey participants were selected using a stratified multi-stage cluster sampling technique within a cross-sectional design. The sampling approach is comprehensively detailed in the SLDHS report [8]. The SLDHS data was available through the World Health Organisation (WHO) Health Equity Assessment Toolkit (HEAT) website [34]. HEAT is a software application developed by the World Health Organisation that facilitates the inspection, analysis, and reporting of health difference data. It includes data from the Health Inequality Data Repository, enabling users to assess and depict health variations effectively. For our research, we utilised disaggregated data from the SLDHS health indicators available in HEAT. It is crucial to acknowledge that HEAT does not encompass all data from the SLDHS; it primarily consists of selected datasets relevant to health equity assessments. This study analysed the FGM dataset within the HEAT to investigate variations in early childhood FGM prevalence in Sierra Leone. The study complied with the protocols specified in the STROBE (Strengthening the Reporting of Observational Studies in Epidemiology) [35].

## Outcome measure and subgroups of variation

The outcome variable for this study was early childhood female genital mutilation and it was derived when women of reproductive age were asked whether they had undergone FGM and, if so, at what age the procedure occurred. The HEAT software binary outcome variable was established by coding responses as '1' if FGM was performed before age five and '0' if FGM was not performed or was performed after age five.

The HEAT software classifies women's economic status into five categories: poorest, poorer, middle-class, rich, and richest. The place of residence was situated in either an urban or rural area. Sierra Leone's sub-national provinces consist of five distinct geographical regions: the Western Area, home to the capital city, Freetown; Northern Province, characterised by its diverse communities and traditional practices; Southern Province, acknowledged for its rich cultural heritage and traditional practices; Eastern Province, marked by unique ethnic groups and cultural traditions; and the Northwest Province, a recent administrative designation encompassing several districts.

## Data analysis

We utilised the online WHO Health Equity Assessment Toolkit to examine health variation data [36]. HEAT provides estimates, confidence intervals, and summary measures of variation, facilitating the assessment of differences in FGM prevalence and permitting informed judgements based on the data. The determinants of variations employed included age, wealth quintiles, education, place of residence, and province. Four measures were utilised to evaluate variation: Difference (D), Ratio (R), Population Attributable Fraction (PAF), and Population Attributable Risk (PAR). D measures the absolute difference in FGM prevalence between two populations, enabling a direct comparison of their respective rates. R assesses the FGM prevalence between two subgroups by calculating the ratio of one subgroup's prevalence to the other, thus offering a relative evaluation. Both difference and ratio are unweighted measures, signifying that they overlook the population sizes of the subgroups and focus solely on the two groups being compared. PAR quantifies the proportion of early childhood FGM cases in the population that can be attributed to a specific variation factor, highlighting its contribution to the overall burden. In contrast, PAF represents the percentage of total FGM prevalence that could potentially be prevented if the influence of this variation factor were entirely removed. These assessments provide insights into the potential impact of reduced variation on overall FGM prevalence. Although the PAR and PAF are conventionally used in cohort studies, they have been extended to cross-sectional studies, especially inequality studies [37,38]. For a comprehensive explanation of the calculation of these measures, please refer to the literature [39]. R and PAF are comparative variation measures employed to evaluate and compare differences among several elements concerning one another. In contrast, D and PAR are definitive measures, offering precise values that quantify the exact difference in FGM prevalence or the proportion of cases attributable to a specific variation factor. This distinction is crucial, as absolute measures such as D and PAR offer explicit insights into the degree of variation, whereas relative measures like R and PAF situate these variations within broader population dynamics. The WHO acknowledged that summary metrics should be displayed in absolute and relative formats to develop policy-relevant conclusions. The summary metrics and calculations of World Health Organisation measures are thoroughly documented in the literature [40,41].

## Ethics approval and consent to participate

This study did not seek ethical clearance since the WHO HEAT software and the dataset are freely available in the public domain.

## Results

### Prevalence of FGM among women who reported undergoing the practice before age five in Sierra Leone, 2008–2019

Fig 1 shows the prevalence of women who had undergone FGM before age five over time in Sierra Leone from 2008 to 2019. The figure shows that the prevalence of women who have undergone FGM before age five declined from 23.2% in 2008 to 12.3% in 2019.

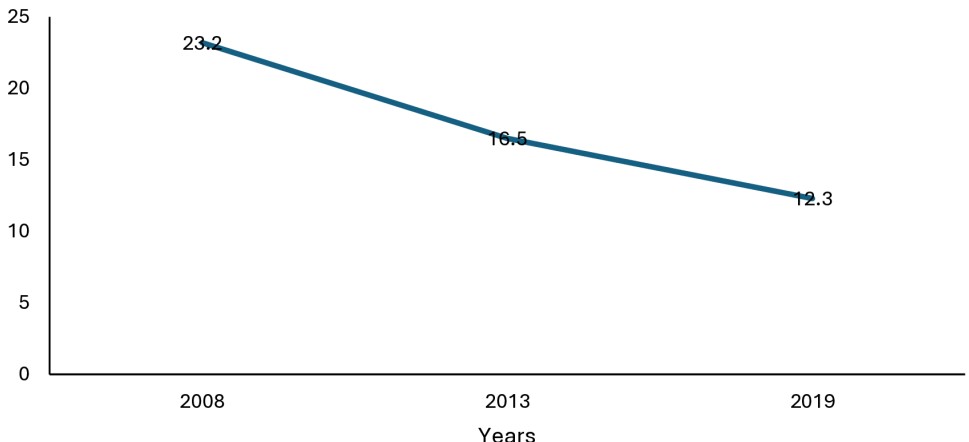

**Fig 1. Prevalence of FGM among women who reported undergoing the practice before five years in Sierra Leone, 2008-2019.**

Table 1 presents the prevalence of women who underwent FGM before the age of five in Sierra Leone over time, categorised by various socioeconomic and geographical factors, from 2008 to 2019. The table indicates an overall decrease in FGM prevalence among women who reported having FGM before age five across different subgroups during this period. Notable variations were observed in the prevalence of FGM among women who reported having FGM before age five across age groups over time. In 2008, women aged 20–29 who reported having FGM before age five had the highest prevalence at 24.5%, while those aged 40–49 had the highest prevalence at 17.2% in 2013. By 2019, women aged 30–39 who reported having FGM showed the highest prevalence at 13%. The lowest prevalence was noted among those aged 15–19 who reported having FGM before age five in 2008 and 2013, at 21.1% and 15.3%, respectively, whereas women aged 20–29 and 40–49 had the lowest prevalence in 2019, each at 12%. In terms of economic status, women in quintile 5 (the richest) consistently reported the lowest prevalence of FGM before age five compared to those in other quintiles. Women in quintile 3 (middle-class) exhibited the highest prevalence in 2008 and 2013, with rates of 26.5% and 18.6%, respectively, while those in quintile 4 (richer) had the highest prevalence in 2019 at 13.1%. Regarding educational attainment, women with no formal education had the highest prevalence of FGM before age five over the years, with rates of 25.7%, 18.4%, and 13.4% in 2008, 2013, and 2019, respectively. Conversely, women with higher education had the lowest prevalence at 10.7% in 2008 and 13.3% in 2013, while those with primary education recorded the lowest prevalence in 2019 at 10.2%. Additionally, women living in rural areas experienced the highest prevalence of FGM before age five in 2008 and 2013, with rates of 25.5% and 17.2%. However, in 2019, urban residents reported the highest prevalence at 12.5%. At the provincial level, women in the Northern province had the highest prevalence of FGM before age five in 2008 at 30% and in 2013 at 21.3%. In 2019, the Northwestern province recorded the highest prevalence at 19.9%.

**Patterns of FGM among women who reported undergoing the practice before age five in Sierra Leone, 2008–2019**

Table 2 shows the variations among women who underwent FGM before the age of five across various subgroups in Sierra Leone from 2008 to 2019. The difference in FGM among women aged 40–49 and those aged 15–19 increased from -3.2 to 0.1 percentage points. The ratio between these two groups increased slightly from 0.9 in 2008 to 1.0 in 2019, indicating no variation. Also, the difference in FGM prevalence among women under five years between those in the richest quintile and those in the poorest quintile decreased from 6.1 percentage points in 2008 to 1.4 percentage points in 2019. The ratio between these groups fell from 1.4 in 2008 to 1.1 in 2019, reflecting a variation that disproportionately affected women in quintile 1. Additionally, the difference in the prevalence of FGM of women under five years old whose mothers had higher education compared with those of mothers with no education decreased from 15 percentage points in

**Table 1.** Prevalence of FGM among women who reported undergoing the practice before age five in Sierra Leone, 2008-2019.

| Subgroup | 2008 | | 2013 | | 2019 | |
|---|---|---|---|---|---|---|
| | Sample | Estimate | Sample | Estimate | Sample | Estimate |
| **Age** | | | | | | |
| 15-19 years | 904 | 21.1 | 2881 | 15.3 | 2095 | 12.1 |
| 20-29 years | 2625 | 24.5 | 5000 | 16.4 | 4533 | 12.0 |
| 30-39 years | 2081 | 21.8 | 4395 | 17.1 | 3832 | 13.0 |
| 40-49 years | 1125 | 24.3 | 2641 | 17.2 | 2471 | 12.0 |
| **Economic status** | | | | | | |
| Quintile 1 (poorest) | 1318 | 23.5 | 2928 | 15.0 | 2473 | 12.7 |
| Quintile 2 | 1302 | 25.3 | 2879 | 16.6 | 2551 | 12.3 |
| Quintile 3 | 1361 | 26.5 | 2979 | 18.6 | 2559 | 12.2 |
| Quintile 4 | 1357 | 23.5 | 3058 | 18.5 | 2703 | 13.1 |
| Quintile 5 (richest) | 1397 | 17.4 | 3074 | 14.1 | 2646 | 11.3 |
| **Education** | | | | | | |
| No education | 4692 | 25.7 | 9002 | 18.4 | 6584 | 13.4 |
| Primary education | 836 | 21.0 | 2035 | 13.4 | 1729 | 10.2 |
| Secondary education | 1087 | 15.5 | 3548 | 13.8 | 4166 | 11.6 |
| Higher education | 119 | 10.7 | 333 | 13.3 | 453 | 11.3 |
| **Place of residence** | | | | | | |
| Rural | 4491 | 25.5 | 10119 | 17.2 | 7458 | 12.2 |
| Urban | 2244 | 18.5 | 4798 | 15.3 | 5474 | 12.5 |
| **Province** | | | | | | |
| Eastern | 1220 | 17.1 | 3299 | 14.9 | 2612 | 7.8 |
| Northern | 2910 | 30.0 | 6056 | 21.3 | 2991 | 14.3 |
| Northwestern | NA | NA | NA | NA | 2329 | 19.9 |
| Southern | 1406 | 18.0 | 3114 | 11.9 | 2152 | 11.2 |
| Western | 1199 | 18.9 | 2448 | 12.8 | 2848 | 9.0 |

NA: Not available

2008 to 2.1 percentage points in 2019. The ratio between these two groups declined from 2.4 in 2008 to 1.2 in 2019, yet variation was evident, disproportionately impacting those of mothers with no education. Furthermore, the difference in the prevalence of FGM among women living in urban areas compared to those in rural areas decreased from 7 percentage points in 2008 to -0.3 percentage points in 2019. The ratio between these groups decreased from 1.4 in 2008 to 1.0 in 2019, showing no variation. Finally, the difference in the prevalence of FGM among women who resided in the Western area compared to those in the Northwestern province decreased from 12.9 percentage points in 2008 to 12.1 percentage points in 2019. The ratio between these two groups increased from 1.8 in 2008 to 2.6 in 2019, indicating variation among the two groups.

## Discussion

This study examined the prevalence and patterns of early childhood FGM in Sierra Leone from 2008 to 2019. The study found a decreasing prevalence of women who have undergone FGM before age five in Sierra Leone, from 23.2% in 2008 to 12.3% in 2019. Women with no education, those in quintiles 3 and 4, who resided in rural areas, and those who lived in North and Northwestern provinces had the highest prevalence of FGM before age 5.

 

**Table 2. Patterns of FGM among women who reported having undergone the practice before five years in Sierra Leone, 2008-2019.**

| Subgroup | Measure | 2008 Estimate (%) | 2008 CI-LB | 2008 CI-UB | 2013 Estimate (%) | 2013 CI-LB | 2013 CI-UB | 2019 Estimate (%) | 2019 CI-LB | 2019 CI-UB |
|---|---|---|---|---|---|---|---|---|---|---|
| Child's Age (15–49 years) | D | -3.2 | NA | NA | -1.9 | NA | NA | 0.1 | NA | NA |
| | PAF | 0.0 | -0.1 | 0.1 | 0.0 | -0.1 | 0.1 | -2.5 | -2.6 | -2.4 |
| | PAR | 0.0 | -2.3 | 2.3 | 0.0 | -1.3 | 1.3 | -0.3 | -1.5 | 0.8 |
| | R | 0.9 | NA | NA | 0.9 | NA | NA | 1.0 | NA | NA |
| Economic Status (Wealth Quintile) | D | 6.1 | NA | NA | 0.9 | NA | NA | 1.4 | NA | NA |
| | PAF | -25.0 | -25.0 | -24.9 | -14.9 | -14.9 | -14.8 | -8.3 | -8.4 | -8.2 |
| | PAR | -5.8 | -7.6 | -4.0 | -2.5 | -3.6 | -1.3 | -1.0 | -2.1 | 0.1 |
| | R | 1.4 | NA | NA | 1.1 | NA | NA | 1.1 | NA | NA |
| Education (4 groups) | D | 15.0 | NA | NA | 5.1 | NA | NA | 2.1 | NA | NA |
| | PAF | -53.9 | -54.1 | -53.7 | -19.4 | -19.7 | -19.2 | -8.3 | -8.5 | -8.0 |
| | PAR | -12.5 | -18.0 | -7.0 | -3.2 | -6.8 | 0.4 | -1.0 | -3.9 | 1.8 |
| | R | 2.4 | NA | NA | 1.4 | NA | NA | 1.2 | NA | NA |
| Place of Residence | D | 7 | NA | NA | 1.9 | NA | NA | -0.3 | NA | NA |
| | PAF | -20.1 | -20.2 | -20.1 | -7.8 | -7.8 | -7.7 | 0.0 | -0.1 | 0.1 |
| | PAR | -4.7 | -6 | -3.3 | -1.3 | -2.1 | -0.4 | 0.0 | -0.7 | 0.7 |
| | R | 1.4 | NA | NA | 1.1 | NA | NA | 1.0 | NA | NA |
| Subnational Province | D | 12.9 | NA | NA | 9.4 | NA | NA | 12.1 | NA | NA |
| | PAF | -26.2 | -26.3 | -26.2 | -28.0 | -28.1 | -27.9 | -36.7 | -36.7 | -36.6 |
| | PAR | -6.1 | -8 | -4.1 | -4.6 | -5.7 | -3.6 | -4.5 | -5.5 | -3.6 |
| | R | 1.8 | NA | NA | 1.8 | NA | NA | 2.6 | NA | NA |

CI-LB: Confidence Interval Lower Bound; CI-UB: Confidence Interval Upper Bound; D: Difference; NA: Not Available; PAF: Population Attributable Fraction; PAR: Population Attributable Risk; R: Ratio

The finding that the prevalence of women under five years who have undergone FGM in Sierra Leone has decreased over time is promising. This decline is attributed to various interventions implemented by the Government of Sierra Leone, in collaboration with health development partners, to discourage the practice. These measures include the signing of a memorandum of understanding in 2012 among eight districts at the time, which prohibited FGM [9], the endorsement of the Child Rights Act of 2007 [42], and the implementation of the National Strategy for the Reduction of Gender-Based Violence [43]. Additionally, Sierra Leone introduced the Sexual Offenses Act in 2012, amended in 2019 [44], and launched the 2020 Gender Equality and Women's Empowerment Policy [45]. Another critical factor that may have contributed to the observed decrease in FGM prevalence among women who reported undergoing FGM before the age of five is the prolonged Ebola outbreak in Sierra Leone between 2014 and 2016. During this period, the practice of FGM was significantly disrupted, and the government implemented a prohibition on FGM as part of broader measures to control the spread of the outbreak [46]. This disruption likely contributed to the decreased prevalence observed in 2019. Despite significant progress in reducing FGM prevalence among children under five, the 12.3% prevalence reported in 2019 in Sierra Leone remains concerning. This indicates that the country is still far from achieving its goal of ending the practice by 2030. On the other hand, given the sensitive socio-cultural nature of FGM, the reported prevalence of women who underwent FGM before the age of five should be interpreted with caution. Such data may be influenced by social desirability bias, stigma, or recall bias, leading to the possibility of both under-reporting and over-reporting of the practice. As FGM is considered a violation of multiple human rights and adversely affects women's physical, mental, and sexual health [47], its continued practice places many women at risk of severe complications and even death. Furthermore, this practice not only leads

to severe health risks but also strains the healthcare system, increases healthcare costs, and disproportionately impacts low-income families. Therefore, efforts to end the practice are desperately recommended.

Consistent with a multicountry study conducted across 12 Sub-Saharan African countries [20], this study found that women with no education had a higher prevalence of FGM and were disproportionately affected compared to their peers with at least a primary education. In Sierra Leone, one key reason is that women with at least a primary education are more likely to understand the dangers of FGM. Educated women are also better equipped to access resources that explain the negative consequences of the practice and work towards ending it. These factors likely contributed to the lower prevalence of FGM among women with at least primary education or higher. However, considering that nearly half (48.4%) of Sierra Leone's population is illiterate [48], the continued high prevalence of FGM poses a significant challenge to efforts by stakeholders to eliminate the practice soon.

This study also revealed that women in the wealthiest quintile (quintile 5) had a lower prevalence of FGM compared to those in poorer quintiles, with women in the poorest quintile (quintile 1) being disproportionately affected. This finding is consistent with a multilevel analysis conducted in Chad, which reported that children aged 0–14 years in the poorer, middle, richer, or richest quintiles had lower odds of undergoing FGM compared to their counterparts in the poorest quintile [20]. Women in wealthier households often have at least a basic education, which increases their awareness of the health risks, human rights violations, and legal consequences associated with FGM. Additionally, wealthier women have greater access to information through platforms such as electronic and print media, which discourage FGM, unlike their counterparts in poorer households who have limited access. Wealthier women also tend to have greater economic independence, reducing their reliance on traditional practices like FGM. Addressing these factors will be essential to ending the practice in Sierra Leone.

Moreover, this study found that women residing in rural areas had a higher prevalence of FGM and were disproportionately affected compared to their counterparts in urban areas. This finding aligns with a study conducted in Tanzania [49], which reported that women aged 0–14 years in rural areas had 2.09 times higher odds of undergoing FGM than those in urban areas. The authors of the Tanzanian study attributed this disparity to limited access to education about the harmful effects of FGM in rural areas and the deep-rooted cultural and traditional practices in these regions. Given that Sierra Leone and Tanzania share similar geographic and cultural characteristics, these reasons may partly explain the variations observed in this study. With 58.7% of Sierra Leone's population residing in rural areas [8], the high prevalence of FGM among women poses a significant challenge to eliminating the practice, as it is deeply entrenched in tradition. To accelerate progress in ending FGM, interventions such as awareness campaigns on the dangers of FGM should prioritise rural areas.

Finally, the study found that women living in the Northern and Northwestern provinces had a higher prevalence of FGM compared to those in the other provinces and were disproportionately affected. Two main reasons may explain this disparity. Firstly, the Northern region had the lowest literacy rate (41%) according to the 2015 Education and Literacy Report [47]. Secondly, before its establishment in 2017, districts in the Northwestern province were part of the Northern province. The Northwestern province is considered the most deprived in Sierra Leone, with limited access to resources to combat FGM. These factors contribute to the disproportionately higher prevalence of FGM in these regions. Addressing these variations will require targeted interventions, including improving access to education and resources in the Northern and Northwestern provinces.

## Implications for policy and practice

Based on the study findings, several critical policy and practice implications emerge for addressing FGM variations in Sierra Leone. The declining trend in early childhood FGM prevalence suggests that existing policies and interventions have been partially effective, but targeted modifications are needed to address persistent variations. Policymakers should prioritise the development of geographically tailored interventions, particularly in the Northwestern province, where the variation ratio has increased to 2.6 compared to the Western area. The reduced but still present educational and

economic variations indicate the need for integrated policy approaches that combine anti-FGM initiatives with broader social development programs, including educational support and economic empowerment for vulnerable communities. Healthcare providers and community workers should be equipped with culturally sensitive training materials and resources that address the needs of less educated and economically disadvantaged populations. Furthermore, the success in reducing urban-rural and age-related variations should inform the scaling up of effective strategies to other subgroups of variation. Local authorities should be empowered to implement context-specific interventions while maintaining alignment with national anti-FGM policies. Additionally, the government should strengthen its monitoring and evaluation frameworks to track progress across all demographic groups, ensuring that interventions remain responsive to changing patterns of variation and that resources are allocated efficiently to areas of greatest need.

## Strengths and limitations

The WHO HEAT online software presents notable strengths and limitations for analysing FGM changes over time and variations in Sierra Leone. As a strength, the software provides a standardised, user-friendly platform that enables comprehensive analysis of health variations through multiple subgroups, offering absolute and relative variation metrics (Difference, Ratio, PAF, and PAR) that enhance the robustness of variation assessment. The software's built-in quality control measures ensure data reliability and comparability across different years and regions, while its ability to generate confidence intervals adds statistical rigour to the findings. However, several limitations warrant consideration. First, the software only includes selected datasets from the Sierra Leone Demographic Health Survey, potentially omitting relevant variables including religion and ethnicity that could provide deeper insights into their role in influencing FGM practices. Second, the pre-defined variation subgroups within HEAT may not capture all contextually relevant factors influencing FGM in Sierra Leone, such as specific cultural beliefs or local customs. Third, the software's reliance on survey data may not fully capture unreported cases or reflect real-time changes in FGM practices. Fourth, the inability to perform advanced statistical analyses, such as multivariate regression, trend analysis, or interaction effects between different variation subgroups due to aggregated datasets, limits the depth of analysis possible. Fifth, our analysis lacks data on ethnicity, religion, and population heterogeneity, which likely influence FGM practices, particularly in regions like the Western and Northwestern Provinces. This limitation underscores the need for caution in interpreting our findings and highlights the importance of future research incorporating these critical factors for a more comprehensive understanding of FGM determinants. Sixth, the reliance on quantitative methods, which, while effective in identifying patterns over time, cannot capture the nuanced complexities or underlying reasons behind these patterns. Complementary qualitative research is needed to explore the sociocultural and contextual factors influencing FGM practices, providing deeper insights to inform targeted interventions and policies. Seven, the software's online nature means that researchers must have consistent internet access, and any technical issues with the platform could impede analysis. Finally, the WHO HEAT tool while valuable for identifying patterns over time, has limitations in capturing health inequities among diverse sub-groups and ethnic minorities. A more critical evaluation of its appropriateness for informing policy decisions is needed, and future research should explore complementary approaches to address these gaps.

## Conclusion

The study reveals a notable decline in early childhood FGM prevalence in Sierra Leone, from 23.2% in 2008 to 12.3% in 2019, demonstrating the positive impact of government interventions. However, persistent variations remain across various socioeconomic subgroups. While variations based on age groups and urban-rural residence have largely diminished, variations persist concerning education, economic status, and geographical location, with the Western-Northwestern provincial divide showing an increase. Therefore, we recommend a multi-faceted approach to FGM prevention that prioritises educational initiatives targeting communities with low educational attainment; and implements poverty reduction strategies in economically disadvantaged areas where FGM prevalence remains high. Also, we recommend developing

province-specific interventions, with particular focus on the Northwestern province; and strengthens community engagement and awareness programs in areas with persistent high prevalence. Furthermore, we recommend and maintaining current successful strategies that have contributed to reducing urban-rural and age-related variations and enhanced monitoring and evaluation systems to track progress and ensure interventions remain effective across all demographic groups. Moreover, future studies should include other socio-economic factors like ethnicity and religion in understanding their role in influencing FGM practices; and those with access to raw data should conduct statistical trend analyses to validate observed patterns and provide stronger evidence for policy and intervention planning.

## Author contributions

**Conceptualization:** Augustus Osborne, Camilla Bangura, Umaru Sesay.

**Data curation:** Augustus Osborne.

**Formal analysis:** Augustus Osborne, Umaru Sesay.

**Investigation:** Umaru Sesay.

**Methodology:** Augustus Osborne, Camilla Bangura.

**Supervision:** Augustus Osborne.

**Validation:** Augustus Osborne.

**Writing – original draft:** Augustus Osborne, Camilla Bangura, Baindu Abu, Umaru Sesay.

**Writing – review & editing:** Augustus Osborne, Camilla Bangura, Baindu Abu, Umaru Sesay.

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
