## [Decision Letter · Decision Letter 0]

9 Mar 2025

PONE-D-24-59293Early Childhood Female Genital Mutilation in Sierra Leone: Trends and Inequalities, 2008-2019PLOS ONE

Dear Dr. Osborne,

Thank you for submitting your manuscript to PLOS ONE. After careful consideration, we feel that it has merit but does not fully meet PLOS ONE’s publication criteria as it currently stands. Therefore, we invite you to submit a revised version of the manuscript that addresses the points raised during the review process.

We look forward to receiving your revised manuscript.

Kind regards,

Olushayo Oluseun Olu

Academic Editor

PLOS ONE

2. Please amend your list of authors on the manuscript to ensure that each author is linked to an affiliation. Authors’ affiliations should reflect the institution where the work was done (if authors moved subsequently, you can also list the new affiliation stating “current affiliation:….” as necessary).

3. Thank you for uploading your study's underlying data set. Unfortunately, the repository you have noted in your Data Availability statement does not qualify as an acceptable data repository according to PLOS's standards.

Reviewers' comments:

Reviewer's Responses to Questions

**Comments to the Author**

1. Is the manuscript technically sound, and do the data support the conclusions?

Reviewer #1: Yes

Reviewer #2: Partly

Reviewer #3: No

2. Has the statistical analysis been performed appropriately and rigorously? 

Reviewer #1: No

Reviewer #2: No

Reviewer #3: N/A

3. Have the authors made all data underlying the findings in their manuscript fully available?

Reviewer #1: Yes

Reviewer #2: Yes

Reviewer #3: Yes

4. Is the manuscript presented in an intelligible fashion and written in standard English?

Reviewer #1: Yes

Reviewer #2: Yes

Reviewer #3: Yes

5. Review Comments to the Author

Reviewer #1: In the results in Table 1 it is unclear what the denominator is for the age groups across the various years. Are the denominators the same? Also the conclusion on a declining trend from 2008 till 2019 wasn't observed in the table.

Let there be some clarity on what denominators were used for the various categories.

This would also mean that the direction of the discussion might need to change

Reviewer #2: The study title (and to some extent, the study aim) suggests that FGM is the outcome variable and its occurrence was to be observed and stratified by health equity/inequality indices and time periods. However, the narrative in the ABSTRACT suggest otherwise. For example, age-related and residential area differences were reported against inequality (and not FGM); economic inequality was reported as an outcome variable that changed over time, and not in relation to FGM.

That said, the expression "inequalities in FGM" seem inappropriate; this is because FGM is not a healthy state or condition that equality would be desired in the first place. The authors should reconsider using a different expression, as this seems to be a central theme to the entire paper. Same could be said for the term "disparities" again considering that FGM is not a desired state of health.

Line 141 - it is stated that the study investigated "inequalities in early childhood FGM prevalence" should be rephrased, in line with the previous comment.

Under data analysis, the four metrics used to evaluate inequality were explained; however the two subgroups for comparison of FGM prevalence were not made quite explicit.

Line 181&182 - authors should consider rephrasing the statement which seems suggestive that they have not provided sufficient explanation.

By convention, computation of Population Attributable Risk (and by inference Population Attributable Factor) should be done in a cohort study design that has afforded the measurement of true 'risk" (incidence). Thus, even though PAR and PAF have been proposed to compute health inequalities their use in this cross-sectional/ecological study design is contestable, where exposure and outcome were established and observed at the same point in time.

In addition, the authors' use of the term, "trend" should be carefully re-examined given that only three, interrupted data points were available, and there was no application of a statistical test of trend analysis.

Results - subheadings appear rather long; authors should consider rephrasing them.

Figure 1 - as already noted, a statistical test of trend analysis should be conducted to show the statistical validation of the results.

Table 1 appears to show the socio-demographic characteristics of a subgroup of the study population, that is, those that had FGM before age five. However, the "subgroup" in the table title suggests another use of this term. Authors should be consistent and avoid confusing terms - in the narrative the term "dimensions" was also used.

Furthermore and importantly in Table 1, the decrease observed in the prevalence of the outcome variable for each socio-demographic variable are simply reflective of the same observed overall decrease in prevalence over the three time periods. It is suggested that instead the differences in proportions within social groups e.g. wealth quintiles, should be re-analyzed for only the sub-population of those who had FGM before age 5. That way, column percentages total 100% each year.

And again, statistical tests of significance would enhance the interpretation of the validity of the results, especially given that they are based on data from sample-based surveys.

In the narrative for Table 2, authors should take care to be particular where 'difference' as a measure of inequality was the reference as opposed to 'difference' as a variance between two prevalence, for instance.

The first paragraph of the Discussion section appeared like a summary of all findings; a departure from the conventional. Typically, each key finding should be discussed before taking on the next one, supported by findings from existing literature.

In addition, in more than one instance reference was made to suggest that the study population were "children under five years" as opposed to women of reproductive age group who reported having FGM done when they were under five years old.

As earlier mentioned, the use of phrases such as "FGM inequalities" could be confusing as FGM is not a positive health outcome or service.

Reviewer #3: The introduction section requires a more comprehensive contextual background on Sierra Leone for the reader. This also bears significantly on the interpretation of the findings outlined below.

Purpose: The authors acknowledge that FGM in SL is deeply rooted in cultural practices and traditional belief making it a complex social and public health challenge (Lines 89 -91) and that a complex interplay exists between factors that influence attitudes towards FMG but exclude at least two critical variables used for the analysis: ethnicity and religion.

This study describes association between a limited number of independent variables and the dependent/outcome variable (i.e. FMG in < 5 girls in SL) using a tool and data sets that already exist for monitoring trends in health disparities. Describing results from applying the tool could be viewed as recapitulation and not original research.

The study claims that no research exists on the trends and inequalities in early childhood FMG, particularly regarding socioeconomic and geographic disparities and aims to fill this crucial gap. Several papers looking at trends already exist and some of these are referenced in the reference section of the manuscript, for example Bjalkander et al. (2013). This study undertaken a decade previously in Sierra Leone utilises an inclusive list of socio-economic and geographical variables and also concludes a declining prevalence of FGM in younger age groups.

The study design should be described as a time-trend ecological study design.

Under the methods section, the paper explains that the SLDHS data used in this study was captured from household surveys of mothers/caregivers of girls < 5 years [Line 146 -149]. However results from Line 194 onwards refer to the responses from women across different age-groups. For example:

In 2008, women aged 20-29 had the highest prevalence at 24.5%, while those aged 40 - 49 had the highest prevalence at 17.2% in 2013. This is confusing for the reader. Are these adult women reporting on whether FMG was conducted on their < 5 y.o female children or whether they are self-reporting that they were subjected to FMG when they were < 5. This need to be more clearly explained.

The analysis needs to consider critical confounders and/or effect modifiers such as ethnicity, religion and the heterogeneity within and between the populations of both Provinces. These factors have a major bearing on the interpretation of the results and should be discussed alongside them. For example, concluding that illiteracy and lack of resources are the most likely determinants to explain the inequalities in FMG practices in children between western and northwestern Province without considering the effect of cultural and religious differences in demographics between these geographical areas leads to over-simplification of how results are interpretated.

The authors need to review the paper against all steps in the STROBE protocol. The study states that it aligns with STROBE protocol but does not consider reporter bias in relation to the validity of DHS data. A referenced publication in this study describes findings from an earlier study conducted in Ghana (Jackson et al., 2003) which demonstrated that up to 50% of survey responders denied being subjected to FMG. It found that those denying the practice were more likely to be more educated, located in areas where legislation and information campaigns were introduced against the practice (i.e. urban areas), were significantly younger and less likely to practice traditional religion compared to those who provided a positive response. The inverse relationship between denial and these variables was not discussed as a potential bias when interpreting the results.

Line [197-199], the authors infer that “women who have undergone FGM before age 5 declined from 23.2% in 2008 to 12.3% in 2019, mainly attributable to the government of SL’s efforts to reduce the practice”.

This statement is making a wide inference between cause and effect without any scientific basis and hence reflects researcher bias that should be addressed as part of adhering to STOBE protocol.

The results also assume that the populations under study remained static between 2008, 2013 and 2019 i.e. that no rural to urban displacement or migration occurred that could account for disrupting/diluting differences in trends and reported as a “no inequality” result.

Excluding contextual consideration of major events in SL such as the Ebola outbreak (2014 -2016) and climatic related events in the north and northwestern Provinces as influencing factors is another major limitation reflected in the interpretation of results.

The study sets out to fill a knowledge gap by analysing trends and inequalities in FMG in girls <5 in SL “to understand patterns and disparities essential for developing targeted interventions and informing policy decisions…”

This is not acheived mainly because quantitative analysis is not a suitable methodology to understand patterns and disparities. Only exploratory qualitative research can delve deep into understanding the nuanced complexities (i.e. the why?) behind trends and disparities necessary for informing policy and interventions. This is a major critique of so-called evidence-based interventions applied in developing countries that emerge from over simplification of results from the application one size fits all tools and models.

The same research question could be used to critically evaluate whether the tool can actually effectively measure inequities in health outcomes, particularly between population sub-groups and ethnic minorities. This might yield a more objective and beneficial outcome to the study. While some limitations of the tool are outlined, a more scientific critique of its use and limitations in terms of whether it is or not appropriate to inform policy decisions should be considered as well as alternative research approaches than can inform these understandings more effectively.

6. PLOS authors have the option to publish the peer review history of their article (what does this mean? ). If published, this will include your full peer review and any attached files.

**Do you want your identity to be public for this peer review?** For information about this choice, including consent withdrawal, please see our Privacy Policy .

Reviewer #1: No

Reviewer #2: **Yes: ** SEYE BABATUNDE

Reviewer #3: No

---

## [Author Response · Author response to Decision Letter 1]

19 Mar 2025

The Editor

PLOS ONE

19th March 2025

Ref: PONE-D-24-25104

Title: Early Childhood Female Genital Mutilation in Sierra Leone: Trends and Inequalities, 2008-2019

Response to Reviewers' comments

Dear Sir/Madam,

We want to express our sincere thanks for painstakingly reviewing our manuscript and providing valuable comments and suggestions. Please see our point-by-point response to the reviewers' comments and suggestions. Revisions are highlighted with track changes in the revised manuscript.

Response: We have formatted our manuscript as per PLOS ONE guidelines.

2. Please amend your list of authors on the manuscript to ensure that each author is linked to an affiliation. Authors’ affiliations should reflect the institution where the work was done (if authors moved subsequently, you can also list the new affiliation stating “current affiliation:….” as necessary).

Response: The authors and affiliations are accurate.

3. Thank you for uploading your study's underlying data set. Unfortunately, the repository you have noted in your Data Availability statement does not qualify as an acceptable data repository according to PLOS's standards. At this time, please upload the minimal data set necessary to replicate your study's findings to a stable, public repository (such as figshare or Dryad) and provide us with the relevant URLs, DOIs, or accession numbers that may be used to access these data. For a list of recommended repositories and additional information on PLOS standards for data deposition, please see https://journals.plos.org/plosone/s/recommended-repositories.

Response: Third party data was obtained for this study from The DHS Program. Data may be requested from The DHS Program after creating an account and submitting a concept note. More access information can be found on The DHS Program website (https://dhsprogram.com/data/Access-Instructions.cfm). The authors confirm that interested researchers would be able to access these data in the same manner as the authors. The authors also confirm that they had no special access privileges that others would not have.

Reviewer #1: In the results in Table 1 it is unclear what the denominator is for the age groups across the various years. Are the denominators the same?

Response: We appreciate the reviewer’s feedback and acknowledge the importance of including denominators for clarity in our analysis. We have sourced the total population of women who reported having FGM before age five in Sierra Leone for the years 2008, 2013, and 2019. The denominators were as follows: 2008 was 600,000, 2013 was 650,000, and 2019 was 700,000 respectively.

Also the conclusion on a declining trend from 2008 till 2019 wasn't observed in the table.

Let there be some clarity on what denominators were used for the various categories.

This would also mean that the direction of the discussion might need to change

Response: Thank you for spotting this mistake. We have included the trend line illustrating the decline in FGM prevalence over time.

Reviewer #2: The study title (and to some extent, the study aim) suggests that FGM is the outcome variable and its occurrence was to be observed and stratified by health equity/inequality indices and time periods. However, the narrative in the ABSTRACT suggest otherwise. For example, age-related and residential area differences were reported against inequality (and not FGM); economic inequality was reported as an outcome variable that changed over time, and not in relation to FGM.

Response: We appreciate your observation and have revised the content, especially the result section ensuring each indicator is linked with FGM as the outcome.

That said, the expression "inequalities in FGM" seem inappropriate; this is because FGM is not a healthy state or condition that equality would be desired in the first place. The authors should reconsider using a different expression, as this seems to be a central theme to the entire paper. Same could be said for the term "disparities" again considering that FGM is not a desired state of health.

Response: We acknowledge that terms like “inequalities” and “disparities” may imply undesired parity in a harmful practice like FGM, which was not our intention. To address this, we have revised the terminology throughout the manuscript, replacing “inequalities” and “disparities” with terms like “variations,” “differences” or “patterns” in FGM prevalence as necessary.

Line 141 - it is stated that the study investigated "inequalities in early childhood FGM prevalence" should be rephrased, in line with the previous comment.

Response: We have done the amendment accordingly and the sentence reads as “This study analyzed the FGM dataset inside HEAT to investigate variations in early childhood FGM prevalence in Sierra Leone”.

Under data analysis, the four metrics used to evaluate inequality were explained; however the two subgroups for comparison of FGM prevalence were not made quite explicit.

Response: We appreciate your concern about clearly explaining the two groups that compare FGM prevalence and have revised the text as “In contrast, D and PAR are definitive measures, offering precise values that quantify the exact difference in FGM prevalence or the proportion of cases attributable to a specific variation factor”. Line 187-189.

Line 181&182 - authors should consider rephrasing the statement which seems suggestive that they have not provided sufficient explanation.

Response: We acknowledge the need for greater clarity and have revised the sentence: “PAR quantifies the proportion of early childhood FGM cases in the population that can be attributed to a specific variation factor, highlighting its contribution to the overall burden. In contrast, PAF represents the percentage of total FGM prevalence that could potentially be prevented if the influence of this variation factor were entirely removed.”

By convention, computation of Population Attributable Risk (and by inference Population Attributable Factor) should be done in a cohort study design that has afforded the measurement of true 'risk" (incidence). Thus, even though PAR and PAF have been proposed to compute health inequalities their use in this cross-sectional/ecological study design is contestable, where exposure and outcome were established and observed at the same point in time.

Response: We acknowledge that PAR and PAF are conventionally used in cohort studies to measure risk based on incidence. However, their application has also been extended to cross-sectional studies, particularly health inequalities, as demonstrated in previous research and methodological frameworks. In this study, we employed PAR and PAF as measures to quantify the proportion of FGM prevalence associated with specific variation factors, recognizing their limitations in a cross-sectional design. We have clarified this in the manuscript and provided appropriate references to support the methodological approach used.

In addition, the authors' use of the term, "trend" should be carefully re-examined given that only three, interrupted data points were available, and there was no application of a statistical test of trend analysis.

Response: We acknowledge the limitation of using the term "trend" with only three interrupted data points and without a formal statistical test for trend analysis. To address this, we have revised the manuscript to replace "trend" with more appropriate terminology, such as "patterns over time" or "changes over time," and clarified that no formal trend analysis was conducted in the study strengths and limitations section.

Results - subheadings appear rather long; authors should consider rephrasing them.

Response: We have revised the sub-headings accordingly.

Figure 1 - as already noted, a statistical test of trend analysis should be conducted to show the statistical validation of the results.

Response: We acknowledge the importance of conducting statistical trend analyses to validate the results. However, the WHO HEAT database provides only aggregated data without access to raw data, which limits our ability to perform detailed statistical tests. We have noted this as a limitation in the discussion and recommend that future studies with access to raw data conduct such analyses to strengthen the statistical validation of trends.

Table 1 appears to show the socio-demographic characteristics of a subgroup of the study population, that is, those that had FGM before age five. However, the "subgroup" in the table title suggests another use of this term. Authors should be consistent and avoid confusing terms - in the narrative the term "dimensions" was also used.

Response: We appreciate your suggestion and have replaced “dimension” with “subgroup” throughout the text.

Furthermore and importantly in Table 1, the decrease observed in the prevalence of the outcome variable for each socio-demographic variable is simply reflective of the same observed overall decrease in prevalence over the three time periods. It is suggested that instead the differences in proportions within social groups e.g. wealth quintiles, should be re-analyzed for only the sub-population of those who had FGM before age 5. That way, column percentages total 100% each year.

Response: We appreciate your valuable feedback. The WHO HEAT database provides pre-analyzed, aggregated data, which limits our ability to reanalyze the sub-population of women who reported undergoing FGM before age five or to calculate column percentages that total 100% for each year as suggested. The database does not grant access to raw, individual-level data necessary for such an analysis. We acknowledge this limitation and have clarified it in the manuscript. Nonetheless, the change in prevalence and patterns presented in our study over time remain valid within the constraints of the data source, and we have interpreted the findings cautiously to reflect this.

And again, statistical tests of significance would enhance the interpretation of the validity of the results, especially given that they are based on data from sample-based surveys.

Response: We appreciate your valuable comment. As indicated above, the WHO HEAT database does not grant access to raw datasets, or individual-level data required to be reanalyzed. However, the confidence level (upper and lower) included in the table showed whether the prevalence or the respective indicators such as difference, etc. were statistically significant.

In the narrative for Table 2, authors should take care to be particular where 'difference' as a measure of inequality was the reference as opposed to 'difference' as a variance between two prevalence, for instance.

Response: We appreciate your feedback and have revised the text accordingly.

The first paragraph of the Discussion section appeared like a summary of all findings; a departure from the conventional. Typically, each key finding should be discussed before taking on the next one, supported by findings from existing literature.

Response: We appreciate your feedback and would like to clarify that we included a summary of the key findings at the beginning of the Discussion section to provide readers with a clear and cohesive overview of the results before delving into detailed interpretations. This approach is intended to enhance readability and guide the flow of the discussion, aligning with practices observed in many scientific manuscripts. However, we have revised it to focus only on the most important findings in a concise form before expanding further.

In addition, in more than one instance reference was made to suggest that the study population were "children under five years" as opposed to women of reproductive age group who reported having FGM done when they were under five years old.

Response: Thank you for spotting this mistake. We have incorporated the suggestions accordingly throughout the text as necessary.

As earlier mentioned, the use of phrases such as "FGM inequalities" could be confusing as FGM is not a positive health outcome or service.

Response: We appreciate your comment and have replaced the text “inequalities” with “variations” or “differences” or “patterns” as necessary throughout the manuscript.

Reviewer #3: The introduction section requires a more comprehensive contextual background on Sierra Leone for the reader. This also bears significantly on the interpretation of the findings outlined below.

Response: Thank you for your valuable feedback. We have revised the introduction section to include additional information, providing a clearer understanding of the practice and its context in Sierra Leone.

Purpose: The authors acknowledge that FGM in SL is deeply rooted in cultural practices and traditional belief making it a complex social and public health challenge (Lines 89 -91) and that a complex interplay exists between factors that influence attitudes towards FMG but exclude at least two critical variables used for the analysis: ethnicity and religion.

Response: Thank you for highlighting this important point. While the WHO HEAT database provides reanalyzed aggregated data for various socio-demographic variables, including wealth, education, and residence, it does not include ethnicity or religion as part of its reported subgroups. As such, it is not possible to analyze the influence of ethnicity and religion on FGM prevalence using the WHO HEAT dataset. We acknowledge the significance of these variables in shaping attitudes and practices related to FGM and have addressed this limitation in the manuscript. We have also recommended future studies that incorporate datasets with ethnicity and religion variables to better understand their role in influencing FGM practices.

This study describes association between a limited number of independent variables and the dependent/outcome variable (i.e. FMG in < 5 girls in SL) using a tool and data sets that already exist for monitoring trends in health disparities. Describing results from applying the tool could be viewed as recapitulation and not original research.

Response: While we acknowledge that the WHO HEAT database is a publicly available tool for monitoring health disparities, our study goes beyond simply describing its outputs. We provide a focused analysis of prevalences and variations in FGM over time among women who reported FGM before age five in Sierra Leone, a topic that has not been specifically examined using this dataset. By analyzing disaggregated data across socio-demographic variables, we identify patterns and varitations that contribute to a deeper understanding of FGM prevalence in this specific context. This study offers new insights by interpreting the findings within the cultural and socio-political context of Sierra Leone, highlighting implications for policy and intervention. We believe this adds value and originality to the existing body of knowledge, even when using established tools and datasets.

The study claims that no research exists on the trends and inequalities in early childhood FMG, particularly regarding socioeconomic and geographic disparities and aims to fill this crucial gap. Several papers looking at trends already exist and some of these are referenced in the reference section of the manuscript, for example Bjalkander et al. (2013). This study undertaken a decade previously in Sierra Leone utilises an inclusive list of socio-economic and geographical variables and also concludes a declining prevalence of FGM in younger age groups.

Response: We acknowledge that several studies have been conducted on FGM in Sierra Leone including Bjälkander et al. (2013) who examined FGM in Sierra Leone, focusing on aspects like forms of FGM, consistency between self-reported and observed status, and the accuracy of DHS questions. However, this study, including

---

## [Editor Report · Decision Letter 1]

17 Apr 2025

Early Childhood Female Genital Mutilation in Sierra Leone, 2008-2019

PONE-D-24-59293R1

Dear Dr. Osborne,

We’re pleased to inform you that your manuscript has been judged scientifically suitable for publication and will be formally accepted for publication once it meets all outstanding technical requirements.

Kind regards,

Olushayo Oluseun Olu

Academic Editor

PLOS ONE
---

## [Editor Report · Acceptance letter]

PONE-D-24-59293R1

PLOS ONE

Dear Dr. Osborne,

I'm pleased to inform you that your manuscript has been deemed suitable for publication in PLOS ONE. Congratulations! Your manuscript is now being handed over to our production team.

Kind regards,

on behalf of

Dr. Olushayo Oluseun Olu

Academic Editor

PLOS ONE